

# A halo of reduced dinoflagellate abundances in and around eelgrass beds

Emily Jacobs-Palmer[1], Ramón Gallego[1,2,3], Ana Ramón-Laca[2,4], Emily Kunselman[5,6], Kelly Cribari[1], Micah Horwith[6] and Ryan P. Kelly[1]

[1] School of Marine and Environmental Affairs, University of Washington, Seattle, WA, USA
[2] Northwest Fisheries Science Center, National Oceanic and Atmospheric Administration, Seattle, WA, USA
[3] NRC Research Associateship Program, The National Academies of Sciences, Engineering, and Medicine, Washington, DC, USA
[4] Ocean Associates, Inc., Arlington, VA, USA
[5] Scripps Institution of Oceanography, University of California, San Diego, La Jolla, CA, USA
[6] Washington State Department of Natural Resources, Olympia, WA, USA

Corresponding author
Emily Jacobs-Palmer, emjp@uw.edu

## ABSTRACT

Seagrass beds provide a variety of ecosystem services, both within and outside the bounds of the habitat itself. Here we use environmental DNA (eDNA) amplicons to analyze a broad cross-section of taxa from ecological communities in and immediately surrounding eelgrass (*Zostera marina*). Sampling seawater along transects extending alongshore outward from eelgrass beds, we demonstrate that eDNA provides meter-scale resolution of communities in the field. We evaluate eDNA abundance indices for 13 major phylogenetic groups of marine and estuarine taxa along these transects, finding highly local changes linked with proximity to *Z. marina* for a diverse group of dinoflagellates, and for no other group of taxa. Eelgrass habitat is consistently associated with dramatic reductions in dinoflagellate abundance both within the contiguous beds and for at least 15 m outside, relative to nearby sites without eelgrass. These results are consistent with the hypothesis that eelgrass-associated communities have allelopathic effects on dinoflagellates, and that these effects can extend in a halo beyond the bounds of the contiguous beds. Because many dinoflagellates are capable of forming harmful algal blooms (HABs) toxic to humans and other animal species, the apparent salutary effect of eelgrass habitat on neighboring waters has important implications for public health as well as shellfish aquaculture and harvesting.

## INTRODUCTION

Seagrass species are ecosystem engineers throughout the world's coastal zones (*Jones, Lawton & Shachak, 1994*), generating and sustaining habitat for a multitude of associated taxa (*Duffy, 2006*). These marine macrophytes also provide a wide variety of essential ecosystem services that directly benefit humans, such as provision of nursery habitat for food species (*Heck, Hays & Orth, 2003*) and coastal protection through sediment accretion and stabilization (*Potouroglou et al., 2017*; reviewed in *Nordlund et al. (2016)*).

Additionally, seagrasses temporarily sequester carbon (*Fourqurean et al., 2012*), and have been predicted to serve as local buffers of pH, particularly for sensitive calcifying taxa under ocean acidification regimes (*Hendriks et al., 2014*; but see *Cyronak et al. (2018)* and *Koweek et al. (2018)*). Seagrass habitat is therefore a globally important resource, with far-reaching positive economic effects.

In addition to these broad ecological and chemical functions, such habitats demonstrate important antimicrobial properties. Seagrass meadows have been shown to reduce exposure to bacterial pathogens affecting humans and marine life, relative to areas lacking such meadows (*Lamb et al., 2017*). Additionally, seagrass tissue itself is a source of diverse secondary metabolites capable of killing bacteria responsible for a variety of serious infections (*Kannan, Arumugam & Anantharaman, 2010*; reviewed in *Zidorn (2016)*). Finally, specific bacteria associated with seagrasses are known to kill or inhibit the growth of taxa that produce harmful algal blooms (HABs) (*Inaba et al., 2017*; reviewed in *Imai (2015)*), which can cause shellfish poisoning, fish kills, and mass de-oxygenation events, among other detrimental effects (reviewed in *Grattan, Holobaugh & Morris (2016)*, *Hallegraeff et al. (2017)* and *Rabalais et al. (2014)*, respectively). Thus, the antimicrobial properties of seagrass and associated organisms also yield benefits for human and local ecosystem health.

Eelgrass (*Zostera marina*) is the dominant seagrass along temperate coasts of the Northern Hemisphere (*Short et al., 2007*). Recent worldwide declines in this species and other seagrass taxa are alarming (*Orth et al., 2006*; but see *Shelton et al. (2017)*), and have been met with local protection measures in some cases, such as designation of seagrass as a "Habitat Area of Particular Concern" (*NOAA Fisheries, 2019*), as well as a "Vital Sign" indicator species (*Puget Sound Partnership, 2019*), and are the object of "no net loss" policies (*NOAA Fisheries, 2014*). Frequently, a tradeoff between eelgrass and aquaculture is presumed when eelgrass habitat and associated conservation efforts compete with shellfish seeding grounds (*Hosack et al., 2006*; *Dumbauld & McCoy, 2015*); depending on the specific practices used to plant and harvest shellfish, such aquaculture can cause significant physical and ecological disturbance to eelgrass beds (*Tallis et al., 2009*). However, bivalves are in fact often proximally associated with seagrass even in the absence of aquaculture, and mutualisms between the two groups have been described (*Peterson & Heck, 2001*; *Van der Heide et al., 2012*, but see *Kelly & Volpe (2007)*). For these reasons, the effects of eelgrass habitat on the abundance of shellfish and associated species are of particular interest in the region of study, and in the many locations worldwide where members of these taxa co-occur.

Given the critical functions of seagrass globally and *Z. marina* eelgrass locally, we aimed to characterize the biological community associated with this ecosystem engineer, comparing the presence and relative abundance of organisms within contiguous beds to points along transects extending outward to bare substrate. Although there is an existing literature on eelgrass communities, such studies often maintain a relatively narrow taxonomic focus and do not make explicit comparisons with immediately adjacent habitat types, perhaps owing to the survey methodology available (*Nelson, 1979*, *1997*). Here we use environmental DNA (eDNA) metabarcoding with universal eukaryotic primers to

quantify *Z. marina*-associated changes across hundreds of taxa simultaneously at five sites and three time-points within the estuarine waters of Washington State, USA. Because of the close ties between shellfish and eelgrass in the study region, we predict that *Z. marina* habitat may modulate the relative abundance of shellfish and/or associated planktonic organisms, in particular, although our methods are capable of identifying significant changes across a diversity of organisms from over a dozen eukaryotic phyla.

# METHODS

## eDNA sample collection

Environmental DNA sequenced at a single genetic locus can provide an assay of community composition consisting of many taxa. The design of the particular PCR primers used largely determines the taxonomic composition, but it is not uncommon to sequence hundreds of taxa from dozens of phyla in a given sampling effort. Here, we targeted a ca. 313 bp fragment of COI using a primer set (*Leray et al., 2013*) known to amplify a broad range of marine taxa including diatoms, dinoflagellates, metazoans, fungi and others; this primer set is broadly used in ecological applications sampling dozens of phyla simultaneously (*Leray & Knowlton, 2015*) as well as those concerned with specific taxonomic groups of interest (*Gibson et al., 2014*).

To determine the biological community composition within *Z. marina* beds and the surrounding habitat from eDNA, we sampled seawater from five sites in Puget Sound: Port Gamble, Case Inlet, Nisqually Reach, Skokomish and Willapa Bay (Fig. 1). We surveyed each location at three timepoints during late spring and summer, in May, July and August of 2017. Specifically, we collected 1 L of seawater using a bleach-cleaned plastic bottle held immediately under the water surface, within 0.6 m of the eelgrass canopy in a total water depth of 0.3–1 m. We collected samples from eelgrass (located at the approximate center of contiguous beds, 47–90 m inside the edge), from each point in a transect extending alongshore at 1, 3, 6, 10 and 15 m outside the edge of the contiguous beds, and from a final location over bare substrate (located 16–670 m outside the edge of the contiguous beds). The bed edge was defined as the point at which shoot density fell below 3 shoots/m$^2$; see Fig. S1 for a transect schematic and Table S1 for precise transect locations by site. All transects ran alongshore, with samples for a given transect gathered at a uniform tidal height (−0.3 to −1 m mean lower low water level). Due to local geography and conditions, it was not always possible to gather all transect samples during each sampling event; a comprehensive list of samples gathered is given in Table S1. We kept samples on ice and in the dark until processing within 4 h by filtering 500 mL from each sample under vacuum pressure through a cellulose acetate filter with 47-mm diameter and 0.45-µm pore size and stored the filter at room temperature in Longmire's buffer (*Renshaw et al., 2015*). The final dataset consisted of 84 water samples.

## Extraction and amplification

To extract DNA from the sample filters, we used a phenol:chloroform:isoamyl alcohol protocol (*Renshaw et al., 2015*). We incubated filter membranes at 65 °C for 30 min before adding 900 µL of phenol:chloroform:isoamyl alcohol and shaking vigorously for 60 s.
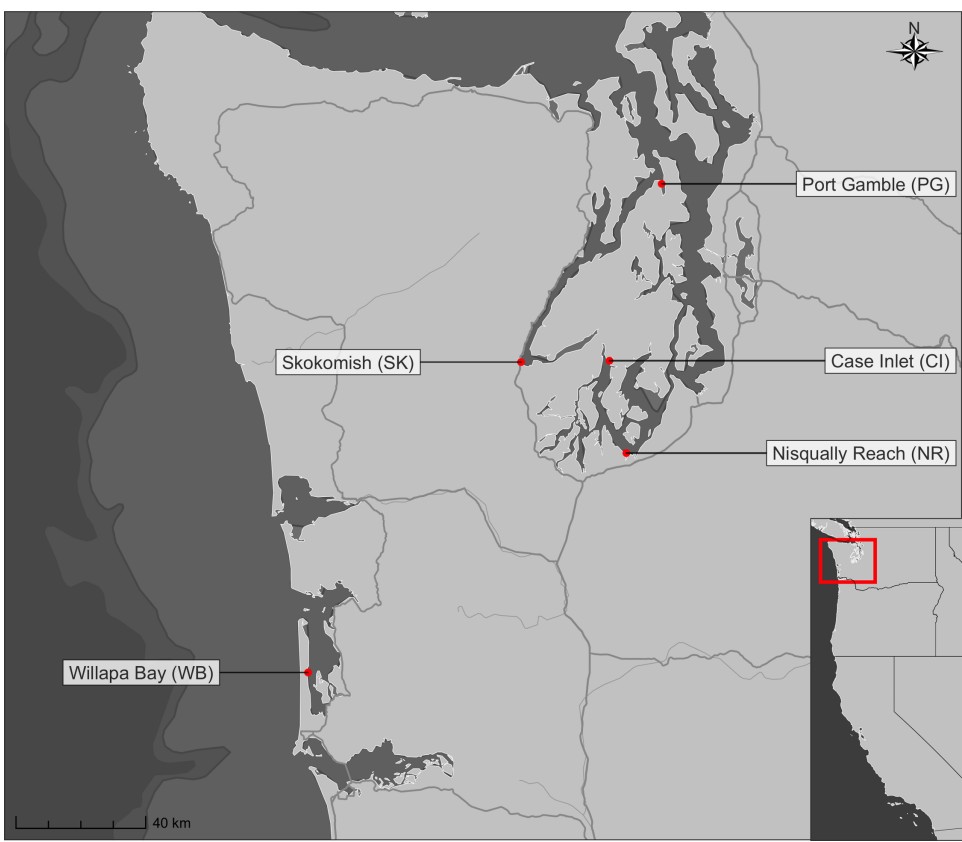

**Figure 1 Nearshore sampling locations in Puget Sound and outer coast, Washington, USA.** GPS coordinates are given in Table S1.

We conducted two consecutive chloroform washes by centrifuging at 14,000 rpm for 5 min, transferring the aqueous layer to 700 μL chloroform, and shaking vigorously for 60 s. After a third centrifugation, 500 μL of the aqueous layer was transferred to tubes containing 20 μL 5 molar NaCl and 500 μL 100% isopropanol, and frozen at −20 °C for approximately 15 h. Finally, all liquid was removed by centrifuging at 14,000 rpm for 10 min, pouring off or pipetting out any remaining liquid, and drying in a vacuum centrifuge at 45 °C for 15 min. We resuspended the eluate in 200 μL water, and used 1 μL of diluted DNA extract (between 1:10 and 1:400) as template for PCR.

To survey the eukaryotic organisms present in our samples, we ran and sequenced in triplicate PCR reactions from each of the 84 biological samples to distinguish technical from biological variance. To sequence many samples and their replicates in a single run while avoiding amplification bias due to index sequence, we followed a two-step PCR protocol (*O'Donnell et al., 2016*). In the first step, we used a PCR reaction containing 1X HotStar Buffer, 2.5 mM MgCl2, 0.5 mM dNTP, 0.3 μM of each primer, and 0.5 units of HotStar Taq (Qiagen Corp., Valencia, CA, USA) per 20 μL reaction. The PCR protocol for this step consisted of 40 cycles, including an annealing touchdown from 62 °C to 46 °C (−1 °C per cycle), followed by 25 cycles at 46 °C. In the second step, following the 2-step PCR protocol given in *O'Donnell et al. (2016)*, we added six base-pair nucleotide

tags to both ends of our amplicons prior to sequencing, allowing us to sequence multiple samples on the same MiSeq run. We allowed for no sequencing error in these tags; only sequences with identical tags on both the forward and reverse read-directions survived quality control. This gave us high confidence in assigning amplicons back to individual field samples.

Finally, we generated amplicons with the same replication scheme for positive controls, comprised of extractions from either kangaroo (genus *Macropus*) or ostrich (genus *Struthio*) tissue. We selected these organisms because they are absent from the sampling sites and common molecular biology reagents, but amplify well with the universal primer set used in this study. Additionally, they can be used to identify possible cross-contamination: reads from other taxa that appear in these positive control samples allow us to estimate and account for the proportion of sequences that are present in the incorrect PCR reaction (see "Bioinformatics" below). We also amplified negative controls (molecular grade water) in triplicate alongside environmental samples and positive controls, and verified by gel electrophoresis that these PCR reactions contained no appreciable amount of DNA.

## Sequencing

To prepare libraries of replicated, indexed samples and positive controls, we followed manufacturers' protocols (KAPA Biosystems, Wilmington, MA, USA; NEXTflex DNA barcodes; BIOO Scientific, Austin, TX, USA). We then performed sequencing on an Illumina MiSeq (250–300 bp, paired-end) platform in four different sets of samples: two MiSeq V.2 runs and two MiSeq V.3 runs. We processed each batch separately through the initial bioinformatics analysis (see below). We employed hierarchical clustering on transects containing six PCR replicates sequenced across two different runs (three technical replicates per run derived from the same sampled bottle of water) and found that these samples were each other's nearest neighbors (Fig. S2); thus sequencing-run-level effects were negligible and we combined the data from the four sequencing runs.

## Bioinformatics

We followed updated versions of previously published procedures for bioinformatics, quality control, and decontamination (*Kelly, Gallego & Jacobs-Palmer, 2018*). This protocol uses a custom Unix-based script (*Gallego, 2019*) calling third-party programs to perform initial quality control on sequence reads from all four runs combined, demultiplexing sequences to their sample of origin and clustering unique variants into amplicon sequence variants (ASVs) (*Martin, 2011*; *Callahan et al., 2016*). The output is a dataset including counts of each ASV per PCR replicate; ~28 million sequence reads from 19370 ASVs emerged from this step.

To address possible cross-sample contamination (*Schnell, Bohmann & Gilbert, 2015*; *Kelly, Gallego & Jacobs-Palmer, 2018*), we subtracted the maximum proportional representation of each environmental ASV across all positive control samples (sequenced from extraction of kangaroo or ostrich tissue) from the respective ASV in field samples; 27 million reads from 19320 ASVs passed this step. After removing the two
PCR replicates with an extremely low number of reads, we estimated the probability of ASV occurrence by performing site-occupancy modeling (*Royle & Link, 2006*; *Lahoz-Monfort, Guillera-Arroita & Tingley, 2016*). Following *Lahoz-Monfort, Guillera-Arroita & Tingley (2016)* and using the full Bayesian code for package rjags (*Plummer, 2013*) provided by those authors, we modeled the probability of occupancy (i.e., true presence) for each of the unique sequence variants in our dataset. We treated replicate PCR reactions of each water bottle as independent trials, estimating the true-positive rate of detection ($P_{11}$), false-positive rate ($P_{10}$) and commonness (psi) in a Bayesian binomial model. We then used these parameters to estimate the overall likelihood of occupancy (true presence) for each ASV; those with low likelihoods (<20%) were deemed unlikely to be truly present in the dataset, and therefore culled. 25 million reads from 3143 ASVs survived this step.

Lastly, we removed samples whose PCR replicates were highly dissimilar: we calculated the Bray–Curtis dissimilarity amongst PCR replicates from the same bottle of water and discarded those with distance to the sample centroid outside a 95% confidence interval. Of 84 bottles of water collected, 3 technical replicates survived QC in 72 cases (86%), two replicates in 9 cases (11%), one replicate in 2 cases (2%), and zero replicates in a single case (1%) (Table S1). The final dataset of 24 million reads from 3142 ASVs comprised 83% of the original sequence reads.

All bioinformatic and analytical code is included in GitHub repositories (*Gallego, 2019*; *Kelly, 2019*), and provides the details of parameter settings in the bioinformatics pipelines used. Sequence and annotation information are included as well, and the former are deposited and publicly available in on the NCBI sequence read archive (SRA accession PRJNA606519).

## Taxonomy

To assign taxonomy to each ASV sequence, we followed the protocol detailed in *Kelly, Gallego & Jacobs-Palmer (2018)*. Briefly, this protocol uses "blastn" (*Camacho et al., 2009*) on a local version of the full NCBI nucleotide database (current as of 13 February 2019), recovering up to 100 hits per query sequence with at least 85% similarity and maximum *e*-values of $10^{-30}$ (culling limit = 5), and reconciling conflicts among matches using the last common ancestor approach implemented in MEGAN 6.4 (*Huson et al., 2016*). Within MEGAN, we imposed an additional more stringent round of quality control to ensure sufficient similarity between query and database sequences by requiring a bit score of at least 450 (ca. 90% identical over the entire 313-bp fragment). Of the 24 million total reads in our dataset, we were able to annotate 4.1 million to the level of phylum or lower; the majority of the remaining reads had no BLAST hits meeting our criteria (7.6 million) or else did not receive taxonomic assignment due to insufficient similarity or conflicting BLAST hits (12.1 million). We use the annotated sequences in our taxonomic analyses below.

Our analysis revealed strong habitat associations for dinoflagellates and not for other taxa (see "Results"). To examine patterns specifically within the phylum Dinoflagellata, we further refined our annotations for these ASVs. Specifically, we considered the geographic

range of taxa involved (restricting possible annotations to those taxa known from the North Pacific) and assigned taxonomy conservatively to the level of family (and genus, when possible) only in cases of >95% sequence identity between the subject and query sequence; ASVs we could not confidently assign to the level of family we excluded from further analyses. Multiple dinoflagellate sequences with identical amino-acid translations occurred within *Heterocapsa*, *Kareniaceae*, *Gymnodinium* and *Hemotodinium*; to avoid pseudoreplication, we treated these as a single taxonomic unit (this choice did not affect the trends or significance of results).

## Statistical analysis

### Community composition

To confirm the spatial resolution of our eDNA communities, we used non-metric multidimensional scaling (nMDS) ordination of eDNA indices for all ASVs within each technical replicate (*Port et al., 2016*). To derive this index, we first normalized taxon-specific ASV counts into proportions within a technical replicate, and then transformed the proportion values such that the maximum across all samples is scaled to 1 for each taxon (*Kelly, Shelton & Gallego, 2019*). Such indexing improves our ability to track trends in abundance of individual taxa in time and space by correcting for both differences in read depth among samples and differences in amplification efficiency among sequences; mathematically, it is equivalent to the Wisconsin double-standardization for community ecology as implemented in the vegan package for R (*Oksanen et al., 2013*). Using this index, we generated a single Bray–Curtis dissimilarity matrix for sequenced transect samples from every unique site/month combination and performed ordinations for each using the metaMDS function of vegan (*Oksanen et al., 2013*; *R Core Team, 2016*) using a maximum of 1,000 random starts. We then created a single Bray–Curtis dissimilarity matrix for our entire dataset and apportioned variance by site, month, transect distance, and sample on the communities present using a PERMANOVA test (implemented with the adonis function in vegan (*Oksanen et al., 2013*)).

### Taxon-Habitat Associations

To examine the relative abundance of phyla in eelgrass habitat relative to bare substrate, we determined eDNA indices for the sum of sequences within each phylum at the two transect extremes (within-eelgrass versus bare), calculating a relative eDNA abundance measure by subtracting the mean eDNA abundance index over bare substrate for each site-month combination from the corresponding mean eDNA abundance index in the eelgrass habitat. Positive values of this measure thus denote higher abundance in eelgrass, while negative values of this index indicate higher abundance over bare substrate. To assess the statistical significance of these phylum-level differences between habitat types, we compared the distributions of mean eDNA abundance indices for individual phyla in samples taken from eelgrass relative to their counterparts taken over bare substrate, using a paired Wilcoxon signed-rank test with Bonferroni correction for multiple comparisons.

### Dinoflagellate distributions

To resolve the fine-scale patterns of dinoflagellates with respect to eelgrass, we focused on transects in which individual dinoflagellate taxa had overall high-abundance. To identify these transects, we took the grand mean of dinoflagellate taxon-specific eDNA indices for each technical replicate along transects at a given time and place, and used the $k$-means function of the R stats package (*R Core Team, 2016*) with $k = 2$ to separate transects from all present dinoflagellate taxa into two groups, high- and low-abundance, using unsupervised machine learning (Fig. S3). Plotting data for individual taxa across transects for each site-month revealed episodic abundance of dinoflagellate sequences in time and space, as expected (Fig. S4). A phylogeny built of dinoflagellate taxa from the high-abundance transects (Fig. S5) confirmed that family- and genus-level taxonomic groups occupied monophyletic clades.

Eight transects identified by unsupervised clustering indicate high-abundance events. For these focal transects, we first compared the eelgrass habitat and bare substrate using a paired Wilcoxon signed-rank test of mean eDNA abundance index for each dinoflagellate taxon (here, having identified sequences to the level of family or genus, rather than grouping dinoflagellates together, as we have done above). Next, to determine whether dinoflagellate abundance measures at intermediate alongshore transect samples (1, 3, 6, 10 and 15 m) were more closely associated with eelgrass habitat or bare substrate, we additionally performed Gaussian mixture modeling with two groups (*Scrucca et al., 2016*). We then used a Wilcoxon rank-sum test to assess the significance of differences in the dinoflagellate eDNA abundance index distribution in the two groups produced by model-based clustering. To ensure that these groups did not result simply from spatial autocorrelation, we calculated Bray–Curtis dissimilarity based on eDNA abundance indices of all ASVs from adjacent points on each full transect. We tested the null hypothesis that spatial distance does not significantly influence Bray–Curtis dissimilarity using a Kruskall–Wallace test (Fig. S6).

## RESULTS

### Community composition

We assigned over 3,000 unique ASVs to 13 eukaryotic phyla comprising a diverse set of single- and multicellular taxa including Arthropoda (arthropods), Annelida (annelid worms), Bacillariophyta (diatoms), Chlorophyta (green algae), Chordata (chordates), Cnidaria (cnidarians), Dinoflagellata (dinoflagellates), Echinodermata (echinoderms), Heterokonta (stramenopiles), Mollusca (molluscs), Nemertea (ribbon worms), Ocrophyta (brown algae) and Rhodophyta (red algae). This represents a broad—although by no means comprehensive—survey of eukaryotic communities in and around our sampled eelgrass beds.

Ordination via nMDS revealed consistent differentiation between eDNA communities across transects within a sampling site and date; technical replicates consistently clustered together. An example plot of samples gathered along the transect from eelgrass to bare substrate at Willapa Bay in July (Fig. 2; all site/date plots shown in Fig. S7) shows that

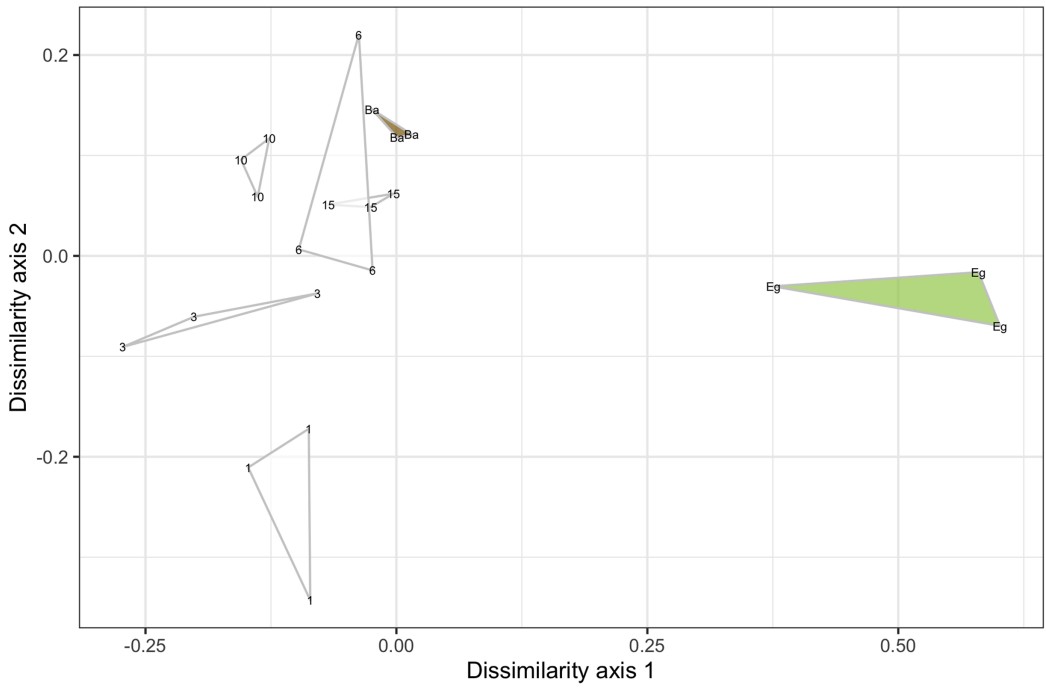

**Figure 2 Example ordination plot of samples along a single transect from bare to eelgrass positions at Willapa Bay in July, 2017.** Technical replicates of each biological sample are grouped as triangles. The sample taken above eelgrass (located 47–90 m inside the edge of the contiguous beds) is shown in green (Eg), alongshore transect samples are shown in white and labeled with distance from the contiguous eelgrass bed in meters, and the sample taken above bare substrate (located 16–670 m outside the edge of the contiguous beds) is shown in brown (Ba).

the eelgrass community is quite dissimilar from other transect points along both axes. Moving away from eelgrass, most technical replicates of each sample bottle are fully distinguishable from those of other sample bottles (non-overlapping in ordination). For the instances in which complete transects were sampled at a given time and place (10) and all three technical replicates of a sample were available for analysis (60), 44 samples (73%) were similarly non-overlapping in ordination with all remaining transect points, demonstrating that despite proximity at the scale of meters, bottles of water contained eDNA evidence of distinct biological communities the majority of the time. Put differently, within-sample variance (reflecting laboratory-driven processes) was smaller than between-sample variance (reflecting biological as well as laboratory processes), hence providing resolution of communities at the scale of meters.

PERMANOVA apportioned the variance in Bray–Curtis distance among samples as follows: site ($R^2$ = 0.186, $p$ = 0.001), month ($R^2$ = 0.079, $p$ = 0.001), and transect distance ($R^2$ = 0.026, $p$ = 0.001) each explain a significant portion of the variance in the dataset. Thus, despite strong effects of location and time, these results confirm that we can consistently distinguish nearshore eDNA communities (as sampled by our primers) at spatial scales of meters for each site and month of sampling. Moreover, we see a highly significant effect of proximity to eelgrass on the complement of organisms present.

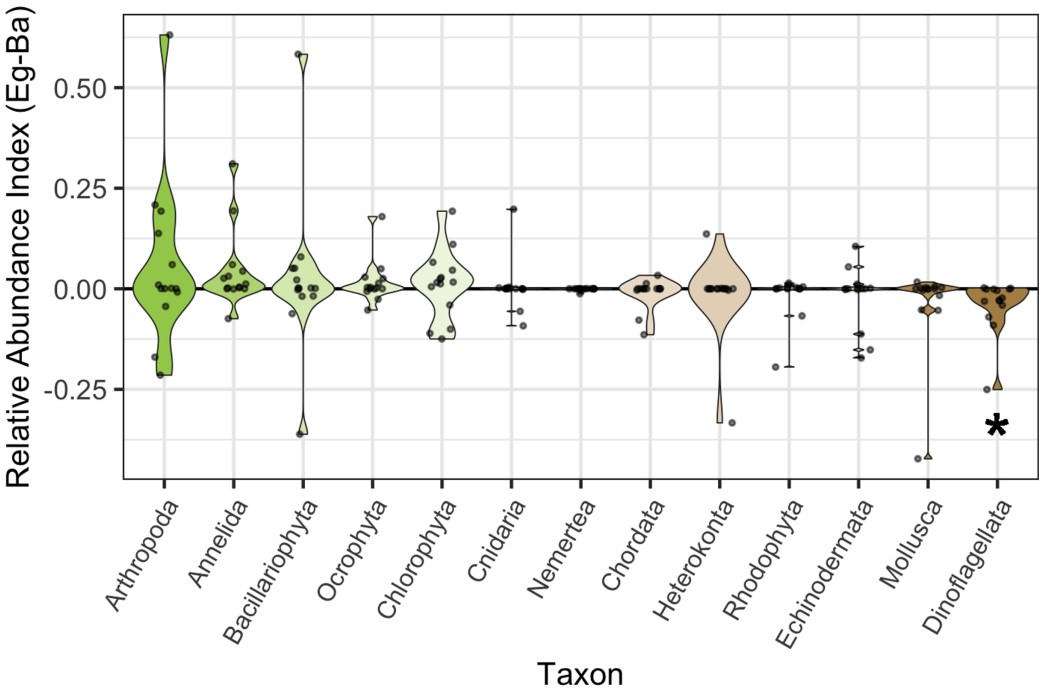

**Figure 3 Habitat associations of sequences assigned to each phylum.** Phyla are ordered and colored by mean relative eDNA abundance index (Eg–Ba: eDNA abundance index in eelgrass minus eDNA abundance index over bare substrate). Greener samples on the left exhibit greater relative abundance in eelgrass, and browner samples on the right exhibit greater relative abundance on bare substrate. The central zero-line indicates no bias in abundance between habitat types. The violins are scaled to a fixed maximum width, and display the density of points within the interquartile range (shown by the violin area). The significance of dinoflagellate habitat associations is denoted with an asterisk (paired Wilcoxon signed-rank test; $p = 0.004$).

## Taxon-Habitat Associations

To determine the habitat preference of major taxa in our dataset at a coarse spatial scale, we classified ASVs to the level of phylum and plotted an index of their relative sequence abundance in eelgrass versus bare positions (Fig. 3). Positive indices denote greater abundance in eelgrass, and negative indices in bare substrate. Across all sites and months, only dinoflagellates show a consistent and strong bias towards one habitat or another; they are nearly universally more abundant over bare substrate. Indeed, the negative association of dinoflagellates with eelgrass beds is the only significant change in phylum-specific abundance between the two habitat extremes after Bonferroni correction for multiple comparisons ($p = 0.004$; paired Wilcoxon signed-rank test). Other single-celled microalgae such as diatoms (Bacillariophyta) and green algae (Chlorophyta) have no significant relationship with eelgrass.

## Dinoflagellate distributions

When dinoflagellates are plentiful relative to background levels, it becomes possible to identify detailed spatial trends in the abundance of individual taxa with respect to eelgrass habitat. To restrict our analysis to such periods, we used unsupervised machine learning (*k*-means clustering) to define a set of high- and low-abundance transects for each

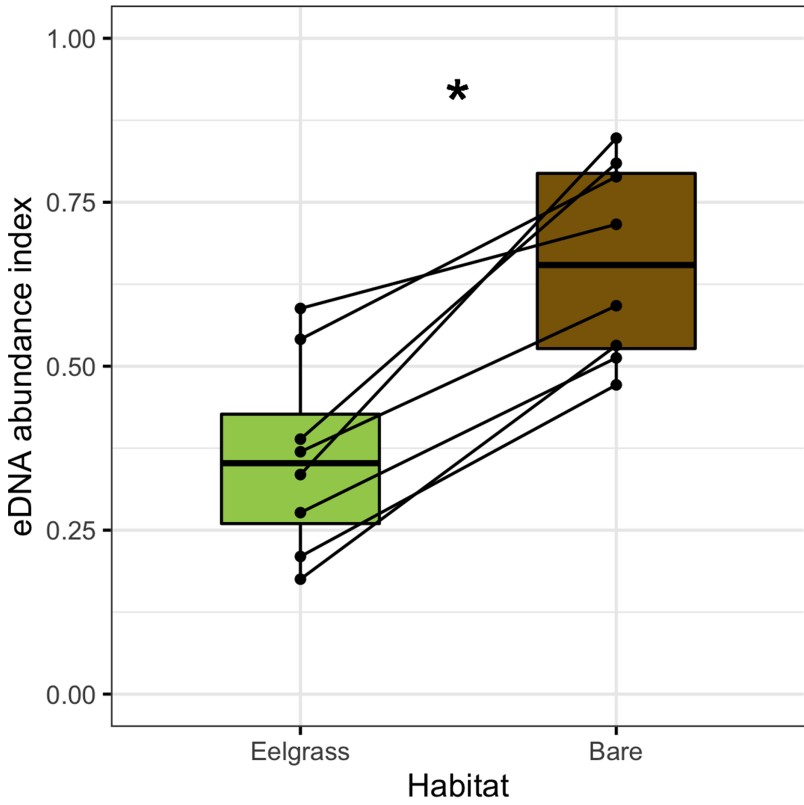

**Figure 4 Habitat preferences of dinoflagellate sequences at site-months in which each taxon occurs at high abundance.** eDNA abundance indices from eelgrass samples are shown in green, and those from bare samples are shown in brown. Each set of points and associated line depict one high-abundance dinoflagellate taxon. The significance of paired differences in eDNA abundance indices for these transect extremes is denoted with an asterisk (Wilcoxon signed-rank test; $p < 0.002$).

dinoflagellate sequence across all sites and months (Fig. S3; between group sum of squares/ total sum of squares = 67.7%); eight transects from eight dinoflagellate taxa appeared in the high-abundance group. Their distributions are indeed highly local and episodic at the scale of our sampling, as expected (Fig. S4).

In this subset of high-abundance transects, the negative interaction of eelgrass and dinoflagellates is taxonomically universal. The taxa represented include two unique variants from the genus *Heterocapsa*, single variants from the genera *Karlodinium*, *Alexandrium*, *Protoceratium* and *Gymnodinium*, and two Kareniaceae family variants of unknown genus, all of which contain known or suspected HAB species (*UNESCO, 2019*). All are also heavily biased towards bare substrate, relative to eelgrass (Fig. 4; Wilcoxon signed-rank test, $p < 0.002$).

After demonstrating a preference of all dinoflagellate taxa towards the bare substrate extreme (when highly-abundant), we then characterized their patterns as a function of distance from the edge of the contiguous eelgrass beds, using data from entire transects (Fig. 5). Examining all points alongshore—and hence, controlling for substrate depth—we found that dinoflagellate eDNA abundance indices at the 1, 3, 6, 10 and 15 m positions

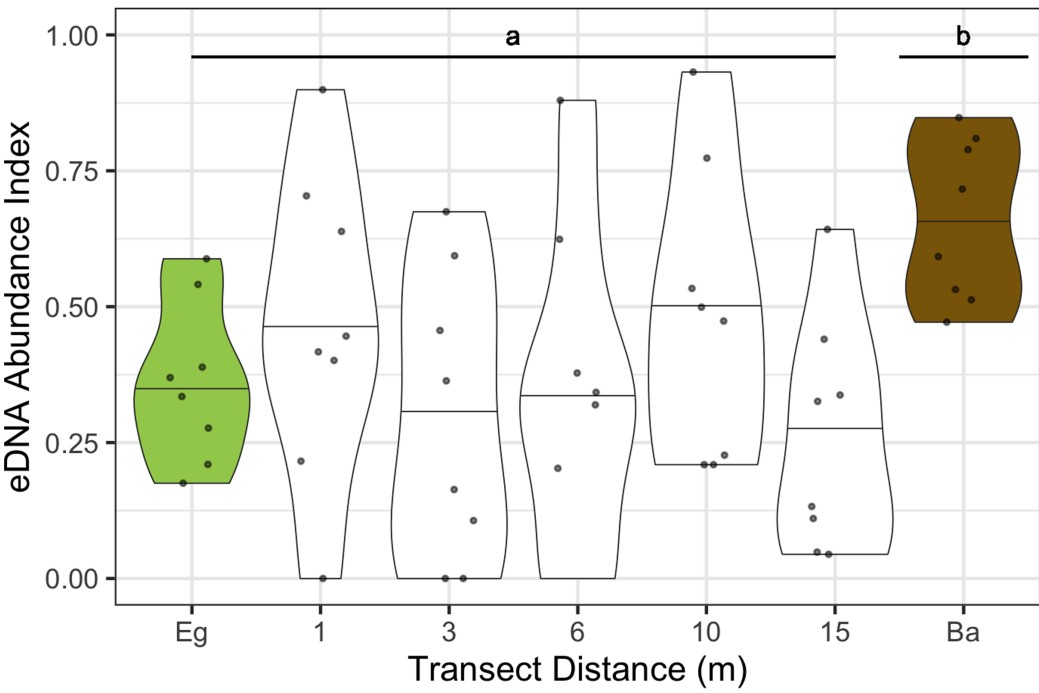

**Figure 5 Dinoflagellate eDNA abundance indices plotted for all sites and months combined at each point along the transect from eelgrass to bare substrate.** The sample taken above eelgrass (located 47–90 m inside the edge of the contiguous beds) is shown in green (Eg), alongshore transect samples are shown in white and labeled with distance from the contiguous eelgrass bed in meters, and the sample taken above bare substrate (located 16–670 m outside the edge of the contiguous beds) is shown in brown (Ba). The violins are scaled to a fixed maximum width, and display the density of points within the interquartile range (shown by the violin area) around the median (central horizontal line). The two clusters produced by model-based clustering and differentiated by Wilcoxon signed-rank test of dinoflagellate eDNA abundance index ($p < 0.02$) are labeled "a" and "b".

grouped with those at the eelgrass position but not the bare position in model-based clustering (Fig. 5 "A" versus "B"). Additionally, the eDNA abundance index of all high-abundance dinoflagellate taxa at these six transect points together differed significantly from bare substrate (Wilcoxon signed-rank test, $p < 0.02$; Fig. 5 "A" versus "B"). These patterns are not simply due to spatial autocorrelation, as overall Bray–Curtis dissimilarity (from all ASVs) shows no pattern associated with geographic distance across full transects (Fig. S6; Kruskall–Wallis rank-sum test, $p > 0.85$).

# DISCUSSION

In a broad-spectrum eDNA survey of the organisms living in and near to eelgrass, we track the relative abundance of a diverse group of taxa from thirteen phyla. We demonstrate the ability of eDNA to distinguish communities represented in samples taken only meters apart, and to reveal a significant axis of variance based on proximity to habitat type, despite strong influences of geography and time across sampling events. One major and significant pattern emerges in our analysis: dinoflagellate taxa are more common over bare substrate than within eelgrass beds when highly-abundant, and this effect extends at
least 15 m beyond the edge of the contiguous beds themselves. Because ours was an observational field study, rather than an experiment, we cannot rigorously distinguish among plausible mechanisms for the observed dinoflagellate distributions. Instead, we use the patterns in our own data as well as the relevant scientific literature to evaluate a number of potential hypotheses.

One plausible mechanism is that of an ecological edge effect acting in nearshore zones at the interface between eelgrass and non-eelgrass habitat. "Edge effects" are changes in the distribution or abundance of a species at a boundary between habitats (defined in *Boström et al. (2011)* and see, for example, *Ries & Sisk (2004)*, *Macreadie et al. (2010)*), or in a broader sense, these same effects across many species, observed at a community level. As patch size in a habitat decreases, edge effects become increasingly important factors influencing the distribution of a species. Here, we observe dinoflagellates at increased abundances in sites with bare substrate 16–670 m away from the edge of contiguous eelgrass beds, and not closer (1–15 m). If dinoflagellates were preferentially living at habitat edges, we would expect higher abundance of these taxa at intermediate distances. If, by contrast, the dinoflagellate pattern arose from a community-wide set of interactions in which eelgrass- and non-eelgrass-associated species overlap at habitat edges, we would expect to see community richness peak at or near the habitat boundary. We observe neither of these patterns in our data (Fig. 5; Fig. S9).

A second plausible mechanism is that slower flow-rates cause plankton deposition within eelgrass beds, but not outside (in parallel with sediment deposition (*Potouroglou et al., 2017*)), such that planktonic cells are recovered at lower levels in our surface water samples. Here, too, we would expect to see a continuum of dinoflagellate abundance as a function of eelgrass thinning with distance from the contiguous beds, and we would expect this to be universal among planktonic species of roughly the same size. Instead, we observe a halo of lowered dinoflagellate abundance even when shoot densities are a few per square meter or lower, and do not see this same pattern for other single-celled algae such as taxa within the groups Chlorophyta or Bacillariophyta. Additionally, we recover eDNA from multiple benthic families (e.g., Dendrasteridae, Tellinidae and Veneridae). It therefore seems unlikely that physical factors driving particle deposition alone produce the observed pattern.

A third possibility is that predatory taxa exist in greater abundance within eelgrass beds and thereby consume dinoflagellates in larger quantities within this habitat (e.g., benthic macrofauna, *Hosack et al., 2006*). This mechanism appears unlikely for two reasons. First, many organisms that eat dinoflagellates also consume diatoms and other single-celled algae; as stated above, we do not observe the same patterns of reduced abundance in and around eelgrass beds for these other taxa. Additionally, a test of habitat associations at the family level reveals no predators of dinoflagellates with significantly higher abundance in eelgrass habitat relative to bare substrate.

We find greater support for the hypothesis that allelopathy from within the eelgrass bed excludes dinoflagellates. A specific allelopathy against microalgal species by *Z. marina* was first described over 30 years ago (*Harrison & Durance, 1985*). More recent evidence suggests this negative interaction applies to multiple HAB taxa that cause paralytic or

diarrhetic shellfish poisoning (including *Alexandrium*, a genus observed in this study), and is mediated locally by a variety of strains of eelgrass-associated algicidal and growth-inhibiting bacteria, particularly from *Erythrobacter*, *Teredinibacter*, *Gaetbulibacter* and *Arthrobacter* genera (*Inaba et al., 2017*) (though the eDNA primers employed here amplify eukaryotes almost exclusively and therefore do not allow us to test this mechanism). However, in our dataset the repressive effect of eelgrass notably does not extend at the phylum level to other phytoplankton such as diatoms (Bacillariophyta) and green algae (Chlorophyta), despite reports that *Z. marina* habitat can deter members of these taxa as well (reviewed in *Gross (2003)*). If we are witnessing patterns associated with an allelopathic interaction between microalgae and eelgrass mediated by bacterial species, it is possible that local variation in the *Z. marina* microbiome (such as that described in *Bengtsson et al. (2017)*) could produce disparate patterns for various microalgal community members. Regardless of mechanism, the taxonomically broad pattern of lowered dinoflagellate abundance within contiguous eelgrass beds and in a halo of influence up to 15 m in radius surrounding the habitat demands explanation.

Dinoflagellates with consistent patterns of abundance-decrease within and around eelgrass habitat in our dataset include species from the genera *Heterocapsa*, *Alexandrium*, *Karlodinium*, *Protoceratium*, *Gymnodinium* and unknown taxa from the Kareniaceae family, each of which have at least one member included in local microscopy-based monitoring programs (*Trainer et al., 2016*; *Kolb, Hannach & Swanson, 2016*); our eDNA methodology thus agrees broadly with previous visual identification of microalgae in this region. Of particular interest are dinoflagellate taxa that include HAB-forming members: the resident species of *Alexandrium* (*A. catanella*) causes paralytic shellfish poisoning via production of saxitoxin (STX; *Wiese et al. (2010)*), species from the genus *Protoceratium* (e.g., *P. reticulatum*) produce yessotoxins (YTXs), whose effects on human consumers of contaminated shellfish are complex and unclear (reviewed in *Tubaro et al. (2010)*), and some species within the family Kareniaceae produce ichthyotoxic karlotoxins *Bachvaroff et al. (2008)*. Saxitoxin, yessotoxins and karlotoxins impact aquaculture and harvest industries directly; detection of STX at concentrations greater than 80 µg STXequiv/100 g is routinely responsible for regional harvest closures (*Moore et al., 2009*), shellfish containing more than 0.1 µg YTX equiv/100 g may not be sold to markets within the European Union, although this toxin is not currently regulated within the US (*Trainer et al., 2013*), and karlotoxins can cause millions of dollars in losses for fisheries in single bloom events (*Hallegraeff et al., 2017*). In summary, the dinoflagellate taxa with low abundance in and around eelgrass habitat in this study have high relevance for local shellfish management decisions, particularly as HABs (including *Alexandrium*) are intensifying with recent ocean warming in the North Pacific (*Gobler et al., 2017*), and are associated with an increase in the number of shellfish harvesting closures (*Trainer et al., 2003*; *Moore et al., 2009*).

## CONCLUSIONS

In order to understand the relationship of *Z. marina* to ecosystem and human health, as well as to shellfish farming and harvest, it is critical to consider our addition of a possible

"action-at-a-distance" element to the existing eelgrass-dinoflagellate interaction model. Given the protected status of *Z. marina* habitat on the Pacific Coast of the United States, the goals of the shellfish industry and eelgrass conservation are often perceived as being in conflict (*Forrest et al., 2009*) and policies prohibit shellfish farming and harvesting within or near beds. For example, in Washington State, required buffer zones between shellfish aquaculture and eelgrass range from 3 to 8 m, depending on the agency involved (*National Marine Fisheries Service West Coast Region, 2017*). However, our work demonstrates that *Z. marina* habitat may have a protective effect against harmful dinoflagellates within these buffer zones, reducing the potential for shellfish to accumulate HAB toxins from the surrounding waters. Likewise, filter feeders can mitigate microbial disease in adjacent environments, and *Magallana* (*Crassostrea*) *gigas*, in particular, has recently been shown to lessen the effects of eelgrass wasting disease on *Z. marina* (*Groner et al., 2018*). Eelgrass and oysters may thus provide critical support to one another in changing marine ecosystems worldwide. Future work will examine their multi-faceted mutualism, characterizing the taxonomic breadth of potential seagrass and shellfish partnerships, as well as defining the molecular mechanisms underlying the roles of both beneficial and detrimental microbial intermediates.

### Funding
This work was supported by the Washington State Department of Natural Resources Interagency Agreement No. 93-096140. The funders had no role in study design, data collection and analysis, decision to publish, or preparation of the manuscript.

### Grant Disclosures
The following grant information was disclosed by the authors:
Washington State Department of Natural Resources Interagency: 93-096140.

### Competing Interests
Ana Ramón-Laca was employed by Ocean Associates, Inc. under contract to National Oceanic and Atmospheric Administration Northwest Fisheries Science Center, Seattle, WA.

### Author Contributions
- Emily Jacobs-Palmer performed the experiments, analyzed the data, prepared figures and/or tables, authored or reviewed drafts of the paper, and approved the final draft.
- Ramón Gallego performed the experiments, analyzed the data, authored or reviewed drafts of the paper, and approved the final draft.
- Ana Ramón-Laca performed the experiments, authored or reviewed drafts of the paper, and approved the final draft.
- Emily Kunselman performed the experiments, authored or reviewed drafts of the paper, and approved the final draft.
- Kelly Cribari performed the experiments, authored or reviewed drafts of the paper, and approved the final draft.
- Micah Horwith conceived and designed the experiments, performed the experiments, authored or reviewed drafts of the paper, and approved the final draft.
- Ryan P. Kelly conceived and designed the experiments, analyzed the data, prepared figures and/or tables, authored or reviewed drafts of the paper, and approved the final draft.

### Field Study Permissions

The following information was supplied relating to field study approvals (i.e., approving body and any reference numbers):

Water only was collected at Washington State Department of Natural Resources remote sites. Consistent with the public trust doctrine, waters of the US are public, and therefore no permit was necessary to conduct this research (see Illinois Central Railroad v. Illinois, 146 U.S. 387 (1892)).

### DNA Deposition

The following information was supplied regarding the deposition of DNA sequences:

The COI sequences are available as a Supplemental File and raw sequence data are available in the Sequence Read Archive (SRA) at accession number PRJNA606519.

### Data Availability

Data and code are available on GitHub at https://github.com/invertdna/EelgrassHalo.

### Supplemental Information

Supplemental information for this article can be found online at http://dx.doi.org/10.7717/peerj.8869#supplemental-information.

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
