# Peer review of "A halo of reduced dinoflagellate abundances in and around eelgrass beds"

_PeerJ, doi:10.7717/peerj.8869_

## Round 0.1 · original submission · Major Revisions

Three expert reviewers have evaluated your manuscript and their comments can be seen below. All of the reviews are favourable, however there are some fundamental issues that need to be corrected as detailed in the reviewers´ comments. If you choose to resubmit your manuscript, please ensure that you address all of these issues in a revision and rebuttal letter.

Reviewer 1 ·

Basic reporting

Line 36 - missing info in place of ?
Line 137 REFREF CITE, c’mon yall
What is happening in lines 237-242, manuscript needs careful attention - I appreciate the R markdown effort but gotta check it over.

Experimental design

I should note I have seen an earlier version of the Jacobs-Palmer paper at another journal; I thought it was sound at the time and required only minor revisions, which the authors provided. I also think this is interesting and useful, and by generating questions will be well cited.


A question on line 67 - you use a fragment of COI and it is “broadly used in ecological applications” but given the focus of this paper, should clarify how those studies have been applied phylogenetically, in other words that those ecological applications have themselves validated use of this marker across the taxonomic breadth considered here, in terms or representing abundance rather than amplicon success.

I greatly appreciate the cross-contamination controls.

Line 154 but surely your data are more comprehensive than known lists of Dinoflagellates in the north Pacific?

Validity of the findings

I believe the findings to be useful and well supported, statistically appropriate. Conclusions are well stated.

Reviewer 2 ·

Basic reporting

-Peer J does not have a requirement in reference style, but there are still some guidelines that should be followed. As a personal preference, I would suggest having the references written on the text, not numbered. Nevertheless, if the authors decide to use numbers on text for the references, then the numbers should go in order, starting at 1, then 2, etc. The first reference cited can not be 22, followed by 8, 18, etc.

-Multiples examples in the text show that the manuscript has not been reviewed properly prior to submission and evince lack of attention to detail: Line 36 “.. ‘Habitat Area of Particular Concern’ (see ? ), line 124, 137, : [REFREF6 CITE],

-The first time an acronym is used, it should be explained in the text. Ex: line 55:” … eelgrass communities can suppress HAB56roducing”. You have never explained the meaning of HAB in the text.

-The introduction has five paragraphs, but only 2 are actually Introduction. Starting at line 44, the authors present the results (which should be moved to Results session, not introduction), and starting at 47 they discuss the results (which should be moved to discussion). The introduction should be rewritten, and only contain information relevant to the introduction.

-The introduction is very short (also 3 out of 5 paragraphs are not introduction-see other comments). The introduction needs more detail and more references to literature. For example, can you expand on the ‘reduced exposure to pathogens”? that is a very relevant topic in your study.

-The whole manuscript would benefit from a review and the rewrite of multiple sessions, to improve language and transitions. Some examples to highlight: lines 33, 34, 244, etc.

-The order of elements in figure caption should be consistent among the different figures. Ex- Fig 2 order is Al, Eg, Ba, while the order in the figure legend is Al, Ba, Eg; Fig3 figure caption is Eg-Ba, etc..

-Figure 1 would benefit from having another figure showing where in the US or world that region on the map is located.

-Figure 2 should be improved. The grey background and the lines make it harder to see the colors, the numbers are very big and are on top of each other. It is also not clear how the triangles were made. If the numbers (or letters) in each edge of the triangle is always the same, I wonder what is the point of having them on the graph. The legend should have a title as such, not FILL. The color brown is not present in the graph but it is present in the legend and in the caption. The legend could actually state what Al, Ba, and Eg mean instead of only having the acronyms.

-Figure 3: The grey background makes it harder to see the colors, also, lines are very tick, and points quite big. I think it would make it more pleasant to see and easier to understand the results if lines were thinners, points smaller, grey background and lines were eliminated.

-Figure 4: The grey background and the lines make it harder to see the colors. I would also eliminate the lines connecting the points. The figure already shows that there is an increase in abundance of dinoflagellates when seagrass is not present, no need for the lines to show the trend for each point.

-Figure 5: the meaning of colors should be stated in each figure. The grey background and the lines make it harder to see the colors, also, lines are very tick, and points quite big. There is no explanation of what areas “a” and “b” mean in the figure.

-What is p11 and p 10, psi? (line 128)

-“Magallana (née Crassostrea) gigas…” what does the neé mean here, I have not seen this notation in scientific papers before.

-“As others had begun to suggest, then…”(337) could you please rephrase this?

Experimental design

-Figure 2 highlights how replicates are variables. Could you explain more why there is only one replicate taken above eelgrass and one from the bare substrate?

-Could the authors describe how the water was collected, and the methodology to guarantee that the samples were not contaminated? Ex- what container was used? Was it acid-washed? Bleached? It is a normal procedure in eDNA sampling to carry a container with distilled/ miliwater as a control to the field, did you have control? How long were the samples kept on ice until processing? Hours? Days? Were they kept in the dark? (time before filtering and amount of light could induce algae growth/ reproduction and alter your results).

-Could you explain better your negative control and the use of kangaroo and ostrich?

-It would be interesting to have replicates bottles for each sampling point and location. It would confirm the results and would show how much a single sampling is actually representative of the diversity richness of the site. Is there a reason why you only collected one little of water in each location per sampling?

-Was there any difference in diversity in the 3 months time point sampled? Do you think that “month” could have affected the results?

Validity of the findings

-I think the study is quite interesting and there is value in reporting the findings of the observational study.

-I like that the authors provided many hypotheses to justify the results (273-296). I would like to see justifications for those hypotheses from the literature. Have other studies suggested some of these hypotheses?


-Fig 3: I assume the points represent the number of identified samples from each one of the categories described in the graph. If that is the case, I find it very suspicions that there is such a clear delimitation where each taxon is only found at a specific distance from eelgrass.

Additional comments

I think the study provides an interesting insight into the effects of eelgrass on dinoflagellates. The manuscript needs some major editing and revision as there are many errors and inconsistencies in the structure and organization of the writing, figures, references in text, etc, which could be an indication of lack of attention by authors and raise concerns about the results.

Reviewer 3 ·

Basic reporting

Generally very well put together. I took this paper on not realising how much of a technical eDNA paper it was but very interested in the idea that eelgrass beds may demonstrate positive environmental impacts. Hopefully the other refree/s is/are technical experts.

One suggestion - on the figures in supplemental tables and figures, it would be easier if the legend for figure S3 was under the first figure and the legend fully explained all aspects of the graph such as the initial representing locations.

Experimental design

I can find few flaws in the technical aspects of this paper (possibly because of ignorance!) and its a really interesting idea that eelgrass beds might have such strong impacts on the water column above them. There has obviously been very close attention to detail in the preparation of this paper the methods and analytical approaches taken appear to be very well described.

Did you sample at a consistent time of day? Was there a long time period between samples on given days?

You measure tidal height in feet??? For consistency I suggest you give this figure in cm or m.

I'd be interested to see time, tidal cycle point, temp, salinity etc also presented in Table S1 and to know that these had been discounted as possible influences,

Validity of the findings

I have a few fundamental questions about this paper (my apologies if they are ridiculous, I'm not expert in the technical aspects of eDNA).

1) You have demonstrated a difference in the presence/absence of dinoflagellate eDNA along transects over eelgrass and bare substrates. Surely it would be much simpler (and cheaper) just to take plankton samples? This would, even if just done for 1 site, validate your results. Could there be some obscure mechanism (e.g. sampling bias) by which eDNA levels are different to abundance of dinoflagellates.

2) Dinoflagellates are true plankton - unable to move against prevailing currents. Current speeds in Puget sound are, apparently, 0.1-0.6 Kts
(https://www.starpath.com/ebooksamples/9780914025160_sample.pdf). I am puzzled as to how any action by dinoflagellates could fight against this so that you have heterogenous distributions over a few meters that can be explained by the presence/absence of eelgrass. I would expect to see patchy distributions whatever the habitats beneath.

3) If the above can be explained and current speed is not a issue, how can you discount the possibility that the difference in distribution is not just a reflection of habitat suitability rather tahn some sort of active exclusion process? Areas above bare sediment may just suit dinoflagellates more than eelgrass habitat and vice versa.

The ordination plots in Supplemental figures generally show that Ba and Eg are closer to each other (CI-May, CI-Aug, PG-May, PG-Aug, Sk-Jul, WB-May, ) than they are to the various points along the transects. In CI-Jul transect points 1,3,6 and 10 are more consistent with Eg than Ba. The high degree of within sample variance surely suggests a lot of noise that could be obscuring any other pattern.

---

## Round 0.2 · Minor Revisions

Below please find an evaluation of your revised manuscript. Please respond to these issues

Reviewer 2 ·

Basic reporting

*Introduction is still too short. I think you could expand more into some topics as the conflict of eelgrass conservation and/or aquaculture, or the services the oyster may depend on provided by the seagrass (line 47) and/or reduced exposure to pathogens.

The authors say: “We have amended the introduction, although it remains brief and contains some results. It was our aim this section to survey the positive effects of seagrass habitats, introduce our study design(including the dominant local seagrass species), and use a very brief statement of results to motivate readers to engage with the study, as appropriate (we did not begin this work expecting to see such strong patterns in Dinoflagellata with respect to eelgrass, and do not wish to write the introduction as such, but do believe that a brief understanding of these patterns will help readers to assess the relevance of the study to their own work). “

I would argue that an introduction should not have the results and that the abstract fits the role the authors are talking about: it is short and provides a very brief statement of results to motivate readers to engage with the study. The introduction should not be short or provide results, it should provide all background information relevant to the study, present results from relevant articles, present your hypothesis and motivation for the study. It should provide more in-depth information on the background of the subject matter. It should also explain your hypothesis, what you attempted to discover, or issues that you wanted to resolve. The introduction will also explain if and why your study is new in the subject field and why it is important.

*I appreciate that the authors re-organized the legends and figures for clarity and consistency

*I think some of the explanation of why using the kangaroo and ostrich sequences could be added to the text

*The authors say” We have added references to two additional studies, including those already mentioned in the text for the first and last hypotheses: “A second plausible mechanism is that slower flow-rates cause plankton deposition within eelgrass beds, but not outside (in parallel with sediment deposition (e.g. Potouroglou 2017)” and “A third possibility is that predatory taxa exist in greater abundance within eelgrass beds and thereby consume dinoflagellates in larger quantities within this habitat (e.g. benthic macrofauna in Hosack 2006).”
It would be interesting to see a discussion of those, as those hypotheses are opposite as one explains why there would be more dinoflagellates within the eelgrass beds, and the other explain why there would be fewer dinoflagellates in the eelgrass beds (they would have been eaten by the predator).

Experimental design

no comment

Validity of the findings

no comment

Additional comments

The authors did a great job addressing the comments made by the reviewers. I added just a few more comments.

---

## Round 0.3 · accepted · Accept

I am satisfied with the changes made to the manuscript.